# Extracting Unlearned Information from LLMs with Activation Steering

**Atakan Seyitoğlu, Aleksei Kuvshinov, Leo Schwinn, Stephan Günnemann**
Department of Computer Science & Munich Data Science Institute
Technical University of Munich
`{a.seyitoglu,a.kuvshinov,l.schwinn,s.guennemann}@tum.de`

## Abstract

An unintended consequence of the vast pretraining of Large Language Models (LLMs) is the verbatim memorization of fragments of their training data, which may contain sensitive or copyrighted information. In recent years, *unlearning* has emerged as a solution to effectively remove sensitive knowledge from models after training. Yet, recent work has shown that supposedly deleted information can still be extracted by malicious actors through various attacks. Still, current attacks retrieve sets of possible candidate generations and are unable to pinpoint the output that contains the actual target information. We propose activation steering as a method for *exact* information retrieval from unlearned LLMs. We introduce a novel approach to generating steering vectors, named Anonymized Activation Steering. Additionally, we develop a simple word frequency method to pinpoint the *correct* answer among a set of candidates when retrieving unlearned information. Our evaluation across multiple unlearning techniques and datasets demonstrates that activation steering successfully recovers general knowledge (e.g., widely known fictional characters) while revealing limitations in retrieving specific information (e.g., details about non-public individuals). Overall, our results demonstrate that *exact* information retrieval from unlearned models is possible, highlighting a severe vulnerability of current unlearning techniques.

## 1 Introduction

Large language Models (LLMs) are trained on vast amounts of text data curated from diverse sources. The extensive training process enables LLMs to generate high-quality responses to a wide range of topics. However, an unintended consequence of this approach is the verbatim memorization of fragments of their training data, which may contain sensitive information. Here, the scale of the training data prevents the reliable identification and removal of all instances of private or copyrighted information, such as personal addresses, passages from copyrighted books, or proprietary code snippets. Regulations, such as the EU's General Data Protection Regulation (GDPR) (Voigt & Von dem Bussche, 2017), aim to give individuals control over their personal information through measures, such as the "Right to erasure", requiring companies to delete user data upon request (Zhang et al., 2023a). However, as retraining large models from scratch is impractical to delete user information, *unlearning* has emerged as an alternative solution to effectively remove knowledge from models while preserving their overall performance (Maini et al., 2024).

Evaluating the success of an unlearning method is a difficult task. Even if the model does not directly respond correctly to questions about the unlearned topic, it doesn't necessarily mean the concept is fully forgotten. Several existing approaches aim to recover unlearned or deleted information (Lynch et al., 2024). Existing works consider an attack successful if the correct answer is present among the candidates but are unable to pinpoint the answer containing the correct information (Patil et al., 2024; Schwinn et al., 2024). In contrast, we argue that for real-world applications, information retrieval systems should incorporate scoring mechanisms that account for the accuracy of selecting the correct answer, which results in a more accurate assessment of leakage risk during deployment.

Our contributions in this paper are as follows:

- We introduce a novel attack based on activation steering (see §2) against LLM unlearning. To this end, we deploy a novel way of generating pairs of prompts to calculate activation differences – Anonymized Activation Steering.

- We evaluate our approach on three unlearning methods and three different datasets. Contrary to existing attacks, we demonstrate that our proposed approach can retrieve unlearned information in an *exact* manner pinpointing the correct answer among a set of candidates with high accuracy.

- We investigate failure scenarios for our approach and find that while *exact* information retrieval is successful on general knowledge (i.e., models unlearned on Harry Potter) it fails for specific, less known information.

- We provide a new dataset based on existing work (Schwinn et al., 2024) for Harry Potter information retrieval, enabling more accurate assessments of unlearning methods.

In this work, we demonstrate the power of activation steering as the evaluation tool for targeted unlearning of LLMs. Our results highlight its effectiveness for broad topics, such as removing copyright-related information, while revealing its limitations when applied to more specific knowledge. By exploring these strengths and constraints, we provide deeper insights into how activation steering affects the behavior of unlearned LLMs, contributing to the development of safe LLMs.

## 2 RELATED WORK

LLM unlearning is a popular research topic due to its efficiency compared to retraining from scratch. Eldan & Russinovich (2023) make a model forget the entire Harry Potter franchise, while Li et al. (2024) unlearn harmful information in biology, chemistry, and cyber-security domains. Apart from unlearning entire domains of information, methods exist that solve the privacy issue of LLMs, deleting personal information at request (Jang et al., 2023; Wu et al., 2023). Meng et al. (2022) replace more specific information from an LLM, unlearning a single sentence at a time.

To effectively test unlearning methods, a clear benchmark is essential. In this context, Maini et al. (2024) introduce the TOFU dataset, which consists of synthetic data about fictitious authors. Because the data is artificial, it allows for direct comparison between a model trained on this dataset that has undergone unlearning, against a baseline model, that is guaranteed to never have seen this dataset during training. Similarly, Li et al. (2024) provide the WMDP dataset, designed specifically for benchmarking unlearning techniques. This dataset contains harmful information, enabling evaluation of a model's ability to effectively forget potentially dangerous content.

Similar to retrieving information from an unlearned model, researchers perform attacks on standard models that refuse to answer harmful prompts. A lot of research in this area is in a "jailbreak" setting where the model already knows the information but refuses to answer (Chu et al., 2024). Shi et al. (2024) identify if a given data was seen during training to recover private information. Attacking an unlearned model is similar, as the goal is to recover information, but in this case, the model does not refuse to answer but outputs seemingly random answers or replaced information such as in unlearning done with ROME method by Meng et al. (2022). Patil et al. (2024) show that ROME and other unlearning methods such as MEMIT (Meng et al., 2023) are prone to attacks, and the information is not truly deleted.

Activation steering is a technique for manipulating LLMs' latent space and guiding the generated responses in the desired direction. This method is applied to overcome refused prompts (Arditi et al., 2024; Rimsky et al., 2024), reduce toxicity (Turner et al., 2023), enhance truthfulness (Li et al., 2023a), and adjust the tone of responses (von Rütte et al., 2024) in LLMs. It works by calculating a "steering vector" that reflects the target direction, derived from pairs of inputs that differ based on the intended direction of behavior (e.g., toxic and non-toxic pairs). The steering vector is obtained through simple subtraction or more advanced techniques like Principal Component Analysis (PCA) or Logistic Regression (Tigges et al., 2023). This vector is then used during generation to steer the model's output towards or away from the desired direction.

# 3 ANONYMIZED ACTIVATION (ANONACT) STEERING

## 3.1 PROBLEM DESCRIPTION

We aim to extract information from an unlearned model about the unlearning topic by asking questions. We ask straightforward questions that have specific keywords as their correct answers. Our objective is to develop a method that functions without prior knowledge of the unlearning topic (such as finetuning the model on a particular area and querying about others) and reliably determines whether the model's response is accurate.

For the base, non-unlearned model, this task of determining the correct answer is trivial, as the most frequent response is typically correct if the model possesses the necessary knowledge in the first place. In contrast, for the unlearned model, information leakage occasionally makes the correct answer appear among the sampled responses, though at a much lower frequency. If the correct answer frequency (CAF) is increased to become the most common response, the user can simply choose the most likely answer, effectively undoing the unlearning process. For this purpose of increasing the CAF, we propose Anonymized Activation (AnonAct) Steering.

## 3.2 ANONACT

We employ a standard activation steering scheme from literature (Arditi et al., 2024; Rimsky et al., 2024) with two distinct types of prompts. Our novel contribution to generating these pairs of prompts is the anonymization of sentences.

For a given question $Q$, we generate multiple anonymized versions. We do it manually for simple entities like character first names and using a large language model, GPT-4, to anonymize more complex terms, like places or institutions. Figure 1 shows an example of this anonymization strategy.

The goal of anonymization is to create prompts close to the original question but without any relation to the unlearned domain. This way, the differences between their activations give us the direction of the unlearned domain. Our method is the first to suggest this type of generalized anonymization to create the difference vectors. Rimsky et al. (2024) use contrastive pairs (such as Yes/No), and Arditi et al. (2024) use completely different prompts to generate the steering vectors.

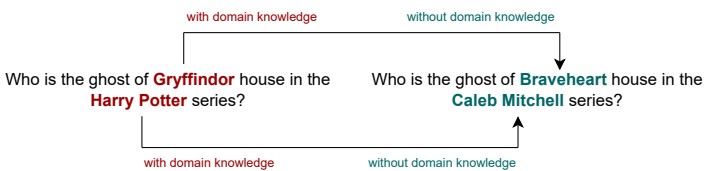

Figure 1: An example anonymization of a question.

After creating the anonymized questions, we follow established methods in the literature for activation steering mainly from Arditi et al. (2024). Specifically, we extract the internal representations between layers of the LLM for each anonymized question, as illustrated in Figure 2. We then compute the difference between these representations and those of the corresponding original questions. By averaging these differences across all anonymized questions, we obtain the steering vector for that layer. During generation, this steering vector is added back with a scaling factor, but only for the generation of the first token. We limit the application of the steering vector to the first token because the internal model representations are captured at that stage. This approach is motivated by findings from previous research, which show that influencing the generation of the first token significantly impacts the entire sequence (Zhang et al., 2023b).

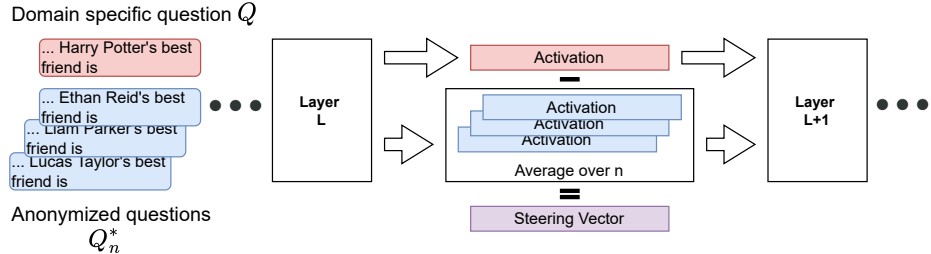

Figure 2: Visual representation for AnonAct.

# 4 EXPERIMENTS

## 4.1 MODELS

For the initial experiments, we use the WhoIsHarryPotter (Eldan & Russinovich, 2023) model, which unlearns all Harry Potter-specific knowledge using sources related to the Harry Potter franchise (books, articles, news posts). It is based on the Llama-2-Chat model, which is a fine-tuned Llama2 (Touvron et al., 2023) for dialog use cases.

Furthermore, to evaluate our approach on other unlearning methods, we use the base model (that was finetuned on the dataset) and codebase authors of TOFU provide to unlearn information from a model. Their model was based on Phi-1.5 (Li et al., 2023b). For ROME (Meng et al., 2022), we use GPT2-XL (Radford et al., 2019) and the code the authors provide to unlearn single facts one by one.

## 4.2 DATASETS

For the Harry Potter unlearning experiments, we curate a dataset of 62 questions using GPT4 by building upon an existing dataset by Schwinn et al. (2024). The dataset is in the format of a Q&A with simple "What" or "Who" questions. We specifically choose questions that are easily answered by a base model (Llama2 7B) to ensure that unlearning is the reason for incorrect answers and not the lack of knowledge in the base model.

For the TOFU dataset (Maini et al., 2024), we utilize 40 questions provided by the authors, focusing on two fictitious authors. These questions are originally designed with open-ended responses. To ensure consistency with our other experiments, we transform this dataset into a keyword-based Q&A format, similar to the Harry Potter dataset, by extracting key terms from the original answers. While the original study uses ROUGE-L score (Lin & Och, 2004) to evaluate answer correctness, we use the extracted keywords to denote an answer as correct.

For ROME Meng et al. (2022), we use 20 keyword-based questions from the CounterFact dataset provided by the authors, for example, about a historical figure's nationality or mother tongue.

To anonymize the questions, we again use GPT4 to generate 5 to 25 suitable replacements for keywords that carry information about the unlearned domain (names, places, fictitious beings) in the questions. If a single question includes multiple anonymized keywords, we use all possible combinations to generate multiple anonymized prompts.

## 4.3 EXPERIMENT SETTING

For the experiments, we follow the work by Arditi et al. (2024) but apply it in an unlearning setting instead of refusal of models. We create the input text using the prompt templates, system prompts, questions, and answer starts, such that the first token to be selected by the model should be the first token of the correct keyword. We compute the internal representations (activations) between layers during the generation of this first token. Then, we calculate the mean activations for the anonymized prompts we generated. Finally, we subtract these mean activations from the activations for the original text to generate the steering vectors (see Figure 2).

We add the steering vectors back during sampling at the generation of the first token only. We use a **Temperature** value of 2 and **Top K** value of 40 to sample 2000 answers for each question. We stop generation at 10 tokens for the Harry Potter and ROME datasets and 50 for the TOFU dataset.

To find the best setting, we ablate different parameters. We use different coefficients for adding the steering vector during sampling, apply our method to different layers (and multiple layers), and finally, use two strategies for generating the steering vector. The first one is **local** calculated as

$$S_l(Q) = A_l(Q) - \frac{1}{N} \sum_{n=1}^{N} A_l(Q_n^*) \tag{1}$$

where $S_l$ is the steering vector for the layer $l$, $A_l$ the activation, $Q$ the given question, and $Q_n^*$ one anonymized version of $Q$ among $N$ anonymized samples. The second setting is called **global**, where we take the mean over all the questions of the local steering vectors and use this mean steering vector for all questions during generation. The global setting requires a dataset of questions and thus is not easily applicable to real-world settings.

After sampling answers using the calculated steering vector, we observe the CAF among the answers generated by the unlearned model without and with our method. We define a "correct" answer as an answer with data leakage. That is, it must include a keyword that shows that unlearned information is leaked. For example, as the answer for "Who is Harry Potter's best male friend?" we accept "Ron." and "Ron Weasley." both as correct, since they both represent data leakage.

To better quantify our results, we calculate the frequencies of words (excluding stop words, which is common practice) and set the probability of an answer being correct to be the maximum frequency value among the words in that answer. We then plot the RoC curve and calculate the AUC score based on these probabilities. An ideal model that only generates correct answers would have an AUC score of 1. If our method pushes the correct keywords to be the most frequent ones, this would also result in a score of 1. This scoring gives us a better indication of performance than just using accuracy or checking if the correct keyword is in a sample of $N$ candidates.

## 5 RESULTS & DISCUSSION

We conduct an initial sampling experiment using the Harry Potter dataset, applying a coefficient of 2 and implementing our method just before the model's final layer. We apply the **local** AnonAct Steering as it more closely aligns with real-world applications. We present the results in Figure 3a. We observe an increase in the CAFs for many questions, with some showing substantial improvements. However, we also acknowledge a slight decrease in performance for a small subset of questions. We note that our objective is to increase the CAFs to the point where they become the most common. For example, a keyword with a low frequency is still the most frequent if no other candidate keyword appears more often. Thus, just looking at the increase in CAFs is insufficient.

To better quantify the success of our method, we employ the simple "most frequent keywords" approach with RoC plots detailed above. We run this experiment in three settings: Base model, unlearned model, and unlearned model using our method. Figure 3b shows the RoC curves for these three settings. We observe that LLAMA2 gets an almost perfect score (0.98). At the same time, the unlearned model has a score of 0.75, which is better than random, showing that information leakage already occurs even without any intervention. Our method sits in the middle of these two but is closer to the base LLAMA2 model with a score of 0.92. The fact that our simple method for determining the correct answer yields a high AUC score indicates that we effectively extract additional information from the model that is supposed to be unlearned.

Next, we conduct the same experiment on the TOFU model and dataset to evaluate how our method generalizes to other unlearning techniques. We assess whether the CAF increases for different unlearning methods. Figure 4 illustrates the outcomes of this experiment, where the CAF is either lower or the same for almost all questions, in contrast to our initial experiments. This shows that our method does not generalize to the TOFU unlearning setting.

To test the generalization of our method further, we use another unlearning method, ROME. We notice that many of the answers in the CounterFact dataset consist of a single token. Therefore, instead

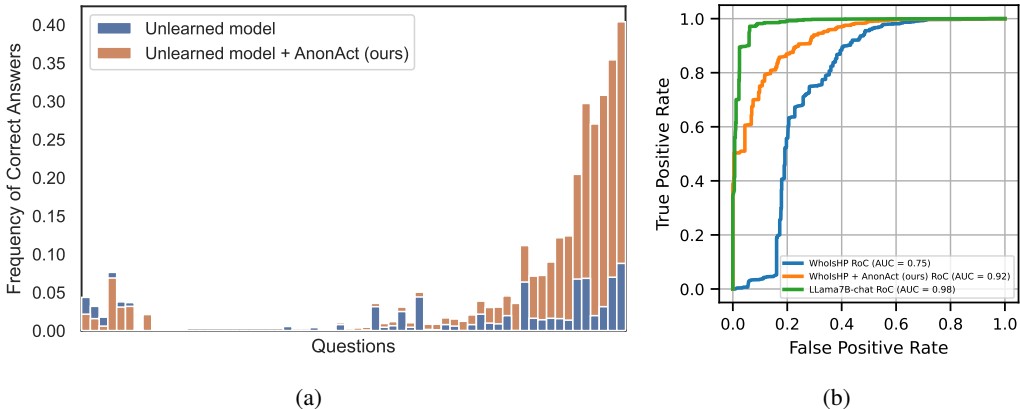

(a)                                          (b)

Figure 3: Experiment results for the Harry Potter dataset. (a) shows the comparison of CAFs between the sampling with an unlearned model without and with using AnonAct. Questions on the x-axis are sorted in ascending order by the difference in the CAF. (b) displays the RoC plots for the base model, unlearned model, and with AnonAct, using keyword frequencies as scores.

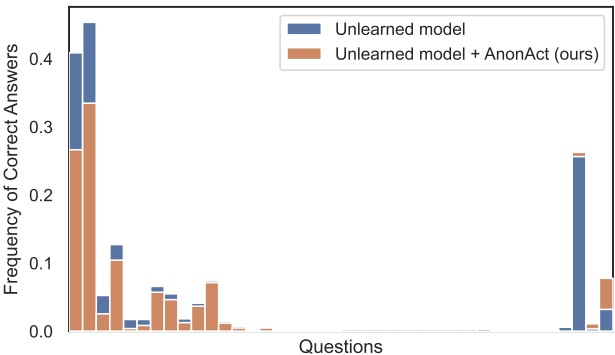

Figure 4: The CAFs from sampling without and with AnonAct for the TOFU experiment. Questions are sorted in ascending order by the increase in performance.

of sampling, we use this subset of questions with single-token answers to plot the probabilities for the top tokens. Figure 5 shows the probabilities of the top 40 tokens for the unlearned model without and with our method for two example questions. Note that ROME is different from the previous two methods, as it replaces the information with another one for unlearning. Our method successfully changes the prediction, leading the sampling away from the forced false token, bringing the distribution to a more uniform state. However, it fails to recover the original true answer: Although the correct answers are in the top tokens for the questions in Figure 5, this is because the given names in the questions are indicative of what the answer might be; not because the model recognizes the subjects.

Our experiments across the three unlearning schemes show that our method is effective in certain cases. We hypothesize that this variation stems from the scope of the subject that was unlearned. The key difference between the successful Harry Potter case and the unsuccessful ones lies in the breadth of the subject matter. Harry Potter represents a large media universe, and unlearning it requires severing connections between its many elements, making it difficult to retrieve related information from any single entity, such as a name. This unlinking effect is effectively mitigated by activation steering with anonymization, allowing the model to restore conceptual associations. In contrast, the TOFU setup involves a single author, and the task is to delete the information about a specific individual. ROME targets even more granular data by removing and replacing a single fact.

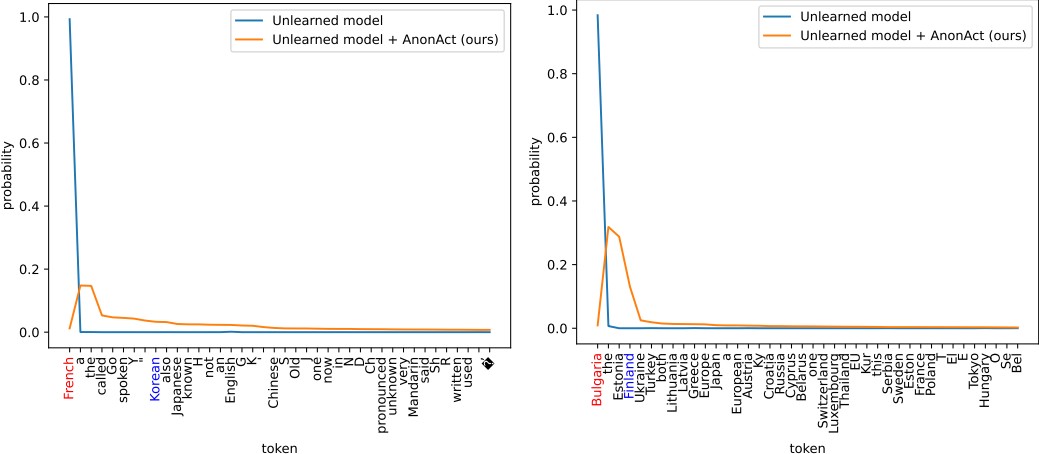

(a) Prompt: "The mother tongue of Go Hyeon-jeong is"    (b) Prompt: "Tapio Kantanen is a citizen of"

Figure 5: Next token probabilities for the completion of the sentences, between the unlearned model and with AnonAct. Blue is the original answer, and red is the replaced one.

While a model follows numerous paths to connect Harry Potter to his best friend–through in-universe concepts like their school, shared adventures, or dialogues, as well as various training data such as books, scripts, articles, and blog posts; the connection in the TOFU case such as between an author and their birthplace, is much more limited. In TOFU, the model learns this connection from a simple sentence about the author's birthplace. Recovering this deleted information by merely steering the model toward the author's direction proves to be ineffective.

## 6 CONCLUSION

In this work, we contribute to information retrieval from unlearned models. To this end, we propose activation steering as a powerful method for this task.

Our novel method employs activation steering in the unlearning context and shows great performance in recovering lost information from broad subjects, such as ones that were unlearned due to copyright. We also show and acknowledge the shortcomings of our method when applied to narrower settings, such as more granular information deletion. We attribute this difference in performance to the way information is unlearned from models. We note that the broader setting, Harry Potter, has many links between the in-universe concepts, and activation steering provides a way to recover these, while narrower topics need a different approach to retrieve unlearned information.

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
