# OpenReview forum: "Extracting Unlearned Information from LLMs with Activation Steering"
_NeurIPS.cc/2024/Workshop/SafeGenAi — SafeGenAi Poster_

### Official Review · Reviewer_uqhJ · 2024-10-09
**Simple and fun**

**Rating:** 6
**Confidence:** 4

**Review:**

### Summary:

This paper presents a method to exploit vulnerabilities in the unlearning process of Large Language Models (LLMs), highlighting the weaknesses of current unlearning techniques. The proposed approach, called Activation Steering, enables precise extraction of information that was supposedly unlearned. By manipulating activation patterns, the authors demonstrate that even when information is removed from a model, it can still be recovered with high accuracy under certain conditions. The study reveals that current unlearning methods are inadequate, as they do not fully eliminate sensitive knowledge, leaving models susceptible to targeted attacks. This work underscores the need for more robust unlearning strategies to prevent unauthorized data retrieval.

### Strengths:
1. **Novelty**: The introduction of activation steering as a technique to retrieve unlearned information is innovative. It advances beyond existing methods by focusing on exact information retrieval, rather than generating sets of possible candidates.

2. **Simplicity**: The method is straightforward, relying on simple activation manipulations and prompt anonymization. This simplicity makes it relatively easy to implement and understand, which enhances its practical applicability.

3. **Direct Identification of Unlearning Issues**: The paper clearly pinpoints the shortcomings of existing unlearning techniques, showing that current methods fail to completely eliminate sensitive data from LLMs.


### Weaknesses:
1.  **Subjective Evaluation Criteria**: The use of keyword frequencies as a proxy for answer correctness could introduce bias and may not fully capture the semantic accuracy of responses.
2. **Lack of tasks**: The paper focuses primarily on unlearning scenarios without exploring how the method might perform on a broader range of tasks. Expanding the experiments to include different types of tasks could provide a more comprehensive evaluation of the method's capabilities.
3. **Unclear Process Description**: The description of the method's process, especially the details of generating steering vectors and the anonymization strategy, is somewhat unclear. More clarity and step-by-step explanation are needed to ensure that readers can easily understand and reproduce the technique.

### Questions
My question is why was the temperature set to 2 in the experiments? Do different temperatures affect the effectiveness of hacking?

### Recommendation
Based on the simplicity of the approach and its effectiveness in clearly identifying the weaknesses in unlearning techniques, I would recommend an **accept**. Addressing the issues of scalability, generalization, and task diversity could further strengthen the impact of this research.

---

### Official Review · Reviewer_hJ1D · 2024-10-09
**Papers points severe evaluation flaw in the Unlearning works**

**Rating:** 6
**Confidence:** 4

**Review:**

In this paper, the authors present an attack to extract supposedly "unlearned" information after unlearning in LLMs.

To achieve this, they rely on the concept of activation steering to guide activations in a certain direction during the LLM inference stage.

In particular, they determine the steering direction by analyzing the activations of the original unlearned prompts and anonymized prompts, then apply the steering direction with a scale factor.

The biggest strength of the paper is that it demonstrates many unlearning techniques still retain knowledge about the unlearned content. This highlights the need for a more robust assessment of unlearning techniques.

Overall, the paper’s findings are important to the LLM unlearning community.